# *Fusarium oxysporum* Casein Kinase 1, a Negative Regulator of the Plasma Membrane H^+^-ATPase Pma1, Is Required for Development and Pathogenicity

**DOI:** 10.3390/jof8121300

**Published:** 2022-12-15

**Authors:** Melani Mariscal, Cristina Miguel-Rojas, Concepción Hera, Tânia R. Fernandes, Antonio Di Pietro

**Affiliations:** Departamento de Genética, Campus de Excelencia Internacional Agroalimentario ceiA3, Universidad de Córdoba, 14014 Cordoba, Spain

**Keywords:** H^+^-ATPase, casein kinase 1, pH, virulence, filamentous fungus, *Fusarium oxysporum*

## Abstract

Like many hemibiotrophic plant pathogens, the root-infecting vascular wilt fungus *Fusarium oxysporum* induces an increase in the pH of the surrounding host tissue. How alkalinization promotes fungal infection is not fully understood, but recent studies point towards the role of cytosolic pH (pH_c_) and mitogen-activated protein kinase (MAPK) signaling. In fungi, pH_c_ is mainly controlled by the essential plasma membrane H^+^-ATPase Pma1. Here we created mutants of *F. oxysporum* lacking casein kinase 1 (Ck1), a known negative regulator of Pma1. We found that the *ck1*Δ mutants have constitutively high Pma1 activity and exhibit reduced alkalinization of the surrounding medium as well as decreased hyphal growth and conidiation. Importantly, the *ck1*Δ mutants exhibit defects in hyphal chemotropism towards plant roots and in pathogenicity on tomato plants. Thus, Ck1 is a key regulator of the development and virulence of *F. oxysporum*.

## 1. Introduction

Fungi can sense and adapt to changes in ambient pH to grow, develop and survive [1,2,3]. Furthermore, these organisms can actively modulate the pH of their surroundings [4,5,6]. For example, plant fungal pathogens are known to alter the pH of the host tissue to promote infection [1,2,3,4,5,6]. While acidification has been described mainly in necrotrophic fungal pathogens, alkalinization has been associated with hemibiotrophs, although this distinction might be less clear-cut since some phytopathogens can trigger both acidification or alkalinization depending on the environment and the infection stage [4]. The root-infecting fungus *Fusarium oxysporum* induces alkalinization of the host during the early stages of infection by secreting a functional homologue of rapid alkalinizing factor (RALF), a family of conserved plant regulatory peptides [6]. An *F. oxysporum* strain lacking F-RALF was unable to induce root alkalinization and showed reduced virulence towards tomato plants [6]. F-RALF appears to target the host receptor-like kinase FERONIA, which in turn prevents the activation of the plant plasma membrane H^+^-ATPase AHA2, causing rapid alkalinization of the apoplast [7].

In fungi, the H^+^-ATPase Pma1 is the most abundant plasma membrane protein and acts as the major regulator of cytosolic pH (pH_c_) [8]. Pma1 is a 100 kDa protein consisting of 10 membrane-embedded domains [9] and a C-terminal autoinhibitory domain [10]. It pumps protons out of the cell and keeps the plasma membrane potential and pH_c_ constant, which is fundamental for cell function and survival [8]. The activity of Pma1 is tightly regulated, mainly at the post-translational level. In the model fungus *Saccharomyces cerevisiae*, glucose activates Pma1 by triggering phosphorylation of the residues S899, S911, and T912, which are highly conserved among fungi [11,12]. Besides glucose, Pma1 is also regulated by ambient pH. For instance, yeast cells treated with acetic acid showed a dramatic increase in *PMA1* transcript levels, which was crucial for acid tolerance [13].

The activity of Pma1 is controlled principally by two kinases, protein kinase 2 (Ptk2), which acts as an activator and casein kinase 1 (Ck1), which functions as an inhibitor. In response to glucose, Ptk2 phosphorylates Pma1 at Ser-899, leading to a 5 to 10-fold increase in H^+^-ATPase activity and acidification of the ambient pH [14,15]. By contrast, Ck1-mediated phosphorylation of Pma1 at Ser-507 inhibits Pma1 activity in *S. cerevisiae* [16]. Ck1 kinase activity, in turn, is controlled by different upstream signaling inputs such target of rapamycin complex 1 (TORC1), protein kinase A (PKA), or oxidative stress [17,18,19]. Besides its role in controlling Pma1 H^+^-ATPase activity, Ck1 may also have a role in sensing alkaline pH by phosphorylating multiple residues of the α-arrestin Rim8, a regulator of the alkaline response Pal/Rim pathway [20].

In *S. cerevisiae,* casein kinase is important for cell morphogenesis, nutrient sensing, and endocytic trafficking [21,22,23]. In the human pathogen *Candida albicans*, the casein kinase Yck2 contributes to the formation of biofilms, cell wall integrity, and virulence [24,25]. Similarly, in the basidiomycete pathogen *Cryptococcus neoformans*, Ck1 was found to regulate cell wall integrity and virulence in a murine systemic infection model [26]. Comparatively few studies have addressed the role of Ck1 in plant pathogens. Recently, *ck1*Δ mutants of the rice blast fungus *Magnaporthe oryzae* were found to have defects in vegetative growth, appressorium formation, and virulence [27], while in *Fusarium graminearum*, deletion of *ck1* caused severe defects in growth, asexual and sexual development and pathogenicity [28]. Here, we examined for the first time the role of Ck1 in pH homeostasis, development, and pathogenicity of a root-infecting plant pathogen. We found that loss of Ck1 in *F. oxysporum* leads to increased Pma1 activity and extracellular acidification as well as a defect in colony growth. Furthermore, we show that Ck1 is crucial for fungal chemotropism towards roots, invasive hyphal growth, and virulence of *F. oxysporum* on tomato plants.

## 2. Materials and Methods

### 2.1. Fungal Isolates and Culture Conditions

*Fusarium oxysporum* f. sp. *lycopersici* wild type (wt) isolate 4287 (FGSC 9935) was used throughout this study. The generation and molecular characterization of *F. oxysporum ck1*Δ mutants and complemented strains was performed, as previously described [29,30]. Fungal strains were stored as microconidial suspensions at −80 °C in 30% (*v/v*) glycerol. For extraction of DNA and generation of microconidia, strains were grown in potato dextrose broth (PDB) supplemented with the appropriate concentration of antibiotics at 28 °C and 170 rpm as described [31].

### 2.2. Generation of Gene Deletion Mutants and Complemented Strains

Targeted deletion of the *ck1* gene (FOXG_05428) was performed by gene replacement with the hygromycin B (Hyg^r^) or phleomycin (Phl^r^) resistant cassette using the split marker method [32]. The sequences of all primers used are listed in Appendix A. Briefly, two PCR fragments encompassing 1.5 kb of the 5′- and 3′-flanking regions were amplified by PCR with the primer pairs Ck1-1 + ck1-3 and Ck1-4 + Ck1-5, respectively. The amplified fragments were then fused to the hygromycin or phleomycin resistance cassettes, previously amplified with primers Gpda15B + Trpter8B, using the fusion primer combinations Ck1-2 + HygG/Phe-5 and Ck1-6 + HygY/LEO, respectively. The two resulting DNA constructs were used to co-transform freshly prepared *F. oxysporum* protoplasts of wt and wt expressing the ratiometric pH probe pHluorin [29]. Obtained transformants were purified by two rounds of monoconidial isolation as described [31]. Deletion mutants were identified first by PCR, and subsequently confirmed by Southern blot (Appendix A). Two µg of genomic DNA of the wt strain and the transformants were treated overnight with the restriction enzyme *Pst*I or *Nsi*I (New England BioLabs, Ipswich, MA, USA) at 37 °C, separated on 0.7% agarose gels, transferred to a nylon membrane and hybridized with the DNA probe indicated in Appendix A, amplified with primers Ck1-2 and Ck1-3.

For complementation of the *ck1* knockout mutant, a DNA fragment encompassing the *ck1* coding region plus 1340 bp 5′- and 1441 bp 3′-flanking sequence was amplified from genomic DNA of the wt strain using the primer pair Ck1-2/Ck1-5. A 2.3 kb fragment with the phleomycin resistance (Phl^r^) cassette was amplified from plasmid pAN8.1 [33] using primers Gpda15B + Trpter8B. Both fragments were used to co-transform the *ck1*Δ #10 mutant as described [34], and phleomycin-resistant transformants passed through two rounds of single conidia isolation were tested for the presence of the *ck1* wild-type allele by PCR with primers Ck1-Fwd and Ck1-Rv.

### 2.3. Colony Growth Assays

For analysis of colony growth, 5 × 10^4^ freshly obtained microconidia of the wt, *ck1*Δ mutant, and *ck1*Δ + *ck1* complemented strains were spot inoculated on yeast extract peptone dextrose (YPGA) or minimal medium agar (MMA) plates. For phenotypical analysis in the presence of stresses, aliquots of 10^5^, 10^4,^ and 10^3^ freshly obtained conidia were spot inoculated on YPGA (1% (*w/v*) yeast extract, 2% (*w/v*) tryptone, 2% (*w/v*) glucose, 2% (*w/v*) agar) medium in the absence or presence of the following compounds at the indicated concentrations: sodium-dodecyl-sulfate (SDS; 7.5 and 12.5 mg/mL), Calcofluor White (CFW; 50 μg/mL), Congo Red (CR; 50 μg/mL), sodium chloride (NaCl; 1.2 M), and potassium chloride (KCl; 1.2 M). Plates were incubated at 28 °C and imaged 3 days after inoculation. All experiments were performed at least twice, with three plates per experiment. To quantify growth, the area of the fungal colony was measured using the MultiGauge software (FujiFilm Corporation, Valhalla, NY, USA). Statistical analysis was conducted using the t-test for unequal variances (Welch’s test).

For analysis of fungal growth in the liquid medium, 5 × 10^6^ microconidia/mL of each *F. oxysporum* strain was inoculated in yeast extract-dextrose (YD) medium buffered at pH 7.4 with 20 mM HEPES (4-(2-hydroxyethyl)-1-piperazineethanesulfonic acid), and 200 µL aliquots were added to wells of a 96-well Microtiter^TM^ Microplate and incubated at 28 °C with shaking at 170 rpm. Fungal growth was evaluated by measuring absorbance at 600 nm in a spectrofluorometer (Infinite M200 PRO, TECAN Life Sciences, Männedorf, Switzerland). Values were normalized to the time-point zero for each strain. All experiments were performed at least twice with triplicate wells for each strain. Statistical analysis was performed on the final time point analyzed using Welch’s test.

### 2.4. Quantification of Microconidia Production and Germination

To measure the rate of microconidial germination, 5 × 10^6^ microconidia/mL of each strain were inoculated in liquid PDB medium and incubated at 28 °C and 170 rpm. After 15 h, the number of germinated conidia was recorded using a Leica DMR microscope. A total of 300 conidia were scored for each condition. Germination was expressed as the percentage of germinated conidia over the total number of scored conidia. To determine the conidiation rate, cultures were inoculated in PDB, as described above, and incubated at 28 °C and 170 rpm for 48 h. The concentration of microconidia was determined by counting a 10^−3^ dilution in a Thoma cell counting chamber under a Leica DMR microscope. Experiments were performed at least twice, with three independent replicates each. Statistical analysis was conducted using Welch’s test.

### 2.5. Measurement of Pma1 H^+^-ATPase Activity

The H^+^-ATPase activity of Pma1 in *F. oxysporum* was measured in the plasma membrane fraction of germlings as previously described [35] with minor modifications [36] by reading absorbance at 750 nm in a spectrofluorometer (Infinite M200 PRO, TECAN Life Sciences, 8708 Männedorf, Switzerland). Specific Pma1 H^+^-ATPase activity was calculated by subtracting the residual activity obtained after the addition of the specific Pma1 inhibitor DES, from the total activity (addition of methanol), expressed in mmol/min/g protein and normalized to time zero. Experiments were performed at least three times, each with three technical replicates.

### 2.6. Cytosolic pH Measurements Using the Ratiometric Fluorescent Probe pHluorin

Measurements of pH_c_ in *F. oxysporum* germlings were performed, as previously described [36], by reading fluorescence emission at 510 nm after excitation at 395 nm and 475 nm in a TECAN spectrofluorometer (Infinite M200 PRO, TECAN Life Sciences, Männedorf, Switzerland). The values of the pHluorin-negative wt background were subtracted for each wavelength, and the 395/475 nm ratio was calculated and converted to pH_c_ values using a pH calibration curve [36]. Experiments were performed at least twice with three independent replicate wells each.

### 2.7. Measurement of Fungal Survival after Acid Treatment

Microconidia of the different strains were germinated for 15 h in PDB at 28 °C and 170 rpm to determine cell survival in extreme acidic conditions. Germlings were resuspended in fresh PDB adjusted to pH 3.0 and incubated 1 h at 28 °C and 170 rpm to allow acids to diffuse through the plasma membrane. After collecting samples for time-point zero (no treatment), either acetic or hydrochloric acid (HCl) was added to a final concentration of 40 mM. After 15, 30, and 60 min, samples were collected, and 5 µL of a 10^−2^ dilution were plated on PDA medium. Plates were incubated 48 h at 28 °C and colonies were counted and normalized to time-point zero. Statistical analysis was conducted using two-way ANOVA and Tukey’s multiple comparisons test. The experiment was performed three times with three technical replicates each.

### 2.8. Measurement of Colony pH

The pH of the medium surrounding the fungal colony was measured as described [37], with minor modifications. Briefly, 5 × 10^4^ freshly obtained microconidia of the different strains were spot-inoculated on MM agar plates. The plates were incubated at 28 °C. At 3, 4, 5, 6, and 7 days after inoculation, samples of 4 mm^2^ were cut either from the center or the margin of the colony and homogenized in 50 μL sterile ultrapure water using a small spatula. The pH of the homogenate was measured using a pH microelectrode (9618S-10D MicroToupH electrode, Horiba). Statistical analysis was conducted using the two-way ANOVA and Tukey’s multiple comparisons test. The experiment was performed three times, with three plates per strain each.

### 2.9. Quantification of Invasive Hyphal Growth and Chemotropism

Invasive hyphal growth through cellophane membranes was determined as described [34], using plates with unbuffered MM or MM buffered with 0.1 M MES (2-(N-morpholino) ethanesulfonic acid) adjusted to pH 5.0 or pH 7.0. Experiments were performed three times, each with three replicates per strain and condition.

Chemotropic growth was measured as previously described using a quantitative plate assay [30]. Briefly, 10^6^ conidia were embedded in 0.5% water agar, incubated 8 h at 28 °C in the presence of a chemoattractant gradient, and the direction of germ tubes relative to a central scoring line was determined in an Olympus binocular microscope at 9.200 × magnification. For pH chemotropism, a gradient competition assay was performed between two wells at both sides of the scoring line containing 25 mM HCl or 25 mM sodium hydroxide (NaOH), respectively, as chemoattractants. For each sample, five independent batches of cells (n = 100 cells per batch) were scored, and three independent plates were prepared for each experimental condition. Calculation of the chemotropic index was done as described [30]. All experiments were performed at least twice.

### 2.10. Western Blot Analysis of MAPK Phosphorylation

Western blot analysis of MAPK phosphorylation levels was performed as described [36,38,39]. Phosphorylated Mpk1 and Fmk1 was detected using rabbit antiPhospho-p44/42 MAPK (Erk1/2) antibody (Thr202/Tyr204; Cell Signaling Technology; #4370), while total Mpk1 and Fmk1 protein was detected with mouse monoclonal anti-Mpk1 antibody (Santa Cruz Biotechnology; sc165979) and a custom-designed polyclonal anti-Fus3 antibody [36]. Mouse anti-α-tubulin antibody (Sigma-Aldrich; #T9026) was used as the loading control.

### 2.11. Plant Infection Assays

Tomato root infection assays were performed in a plant growth chamber (15 h:9 h, light:dark cycle, 28 °C) as described [31]. Ten plants were used per treatment. Plant survival was recorded daily for 40 days as described [34] by the Kaplan–Meier method and compared among strains using the log-rank test. The experiment was performed twice.

## 3. Results

### 3.1. Generation of ck1 Deletion Mutants

A BLASTp search of the F. oxysporum protein database with the S. cerevisiae Yck1 (YHR135C) and Yck2 (YNL154C) amino acid sequences identified a single Ck1 orthologue, FOXG_05428, that encodes a predicted 453 amino acid protein showing approximately 70% identity with Yck1 and Yck2 and 85% identity with Ck1 from C. neoformans (Appendix A).

To examine the role of Ck1 in *F. oxysporum,* we generated *ck1*∆ mutants by replacing the entire *FOXG_05428* coding region with the hygromycin resistance cassette (Appendix A). Southern blot analysis of six independent transformants revealed two *ck1*Δ mutants (*ck1*Δ#10 and #22), in which the 6.1 kb hybridizing *Pst*I fragment corresponding to the wt *ck1* allele had been replaced by a single hybridizing fragment of 3.4 kb, consistent with one homologous insertion of the deletion construct (Appendix A). We also identified a transformant carrying an ectopic insertion of the deletion construct (*ect*#70).

Next, we complemented the mutant strain *ck1*∆#10 by co-transforming protoplasts with the phleomycin resistance cassette together with a 4.3 kb PCR product encompassing the complete *ck1* wt gene. PCR analysis with gene-specific primers identified eleven independent transformants showing a PCR amplification product identical to that obtained from the wt strain, which was absent in the *ck1*∆ mutants, suggesting that these *ck1*∆ + *ck1* transformants had integrated the wt *ck1* allele (Appendix A).

### 3.2. Ck1 Controls Vegetative Growth and Conidiation of F. oxysporum

The *ck1*Δ mutant strains exhibited a marked (approximately 80%) reduction in colony growth compared to the wt, the ectopic transformant, and the complemented strains, both on complete and minimal medium plates (Figure 1 and Appendix A). The growth rate of the *ck1*Δ mutant was also reduced in liquid YD medium, as determined by absorbance at 600 nm (Figure 2A), although the germination rate of the microconidia was not affected (Figure 2B). Furthermore, the production of microconidia in the *ck1*Δ mutant was reduced by 60% compared to the wt and complemented strains (Figure 2C).

### 3.3. Ck1 Is Required for Tolerance to Membrane and Hyperosmotic Stress

We next tested sensitivity to different stresses and found that the *ck1*Δ mutant was more sensitive to membrane and osmotic stress caused by SDS or by high concentrations of NaCl or KCl, respectively (Figure 3). By contrast, the sensitivity of the *ck1*Δ mutant to the cell-wall perturbing agents Calcofluor white (CFW) and Congo Red (CR) was not altered (Appendix A).

### 3.4. Ck1 Acts as a Negative Regulator of Pma1 Activity and Controls Both Extra- and Intracellular pH Homeostasis

In *S. cerevisiae*, Ck1 negatively regulates the H^+^-ATPase activity of Pma1 by phosphorylating the Ser-507 residue [16]. Inspection of an amino acid sequence alignment between Pma1 proteins from yeast and different filamentous fungi, including *F. oxysporum* revealed that this region is highly conserved, with a conserved Thr residue at the analogous position (Appendix A) and a consensus phosphorylation motif for Ck1 [40,41]. This suggested that the regulation of Pma1 by Ck1 may be functionally conserved in *F. oxysporum.* Indeed, we found that Pma1-associated H^+^-ATPase activity in the *ck1*Δ mutant was approximately 3-fold higher than the wt and complemented strain (Figure 4A).

We next asked whether the increased Pma1 activity in the *ck1*Δ mutant leads to the accumulation of extracellular H^+^ and concomitant acidification. While the pH of the wt colonies increased steadily from around 6.0 on day 3 to 6.7 on day 7, the pH in the *ck1*Δ mutant colonies was significantly lower (5.7) and remained stable throughout the monitored time period (Figure 4B). The complemented strain showed an intermediate extracellular pH value.

To study how ck1 deletion affects pH_c_, we introduced the gene knockout construct, with the phleomycin resistance cassette, into a wt background expressing the fluorescent ratiometric pH_c_ sensor pHluorin [36]. PCR analysis identified six phleomycin-resistant transformants producing amplification patterns indicative of a homologous replacement of the *ck1* gene. Southern blot analysis of genomic DNA confirmed the replacement of a 4.1 kb NsiI hybridizing fragment corresponding to the wt ck1 allele, by a hybridizing fragment of 2.5 kb, consistent with homologous insertion of the deletion construct at the ck1 locus (Appendix A). We next used ratiometric pHluorin-based measurements [36] to compare pH_c_ in the wt and the *ck1*Δ mutant. We found that the homeostatic pH_c_ of the *ck1*Δ mutant under the conditions studied was around 6.1, which is approximately 0.7 pH units lower than the pH_c_ of the wt strain (Figure 4C).

We reasoned that hyperactivation of Pma1 H^+^-ATPase activity in the *ck1*Δ mutant results in an increased export of H^+^ and may thus lead to an enhanced tolerance to cytosolic acidification triggered by acetic acid (AA). At acidic ambient pH, AA is present in its undissociated form which is soluble in lipids. Once inside the cell, at close to neutral pH, AA dissociates, resulting in an increased release of protons [42]. We found that addition of AA to the wt strain led to a marked and irreversible decrease of pH_c_ by approximately one pH unit (Figure 5A). We next tested the survival of the cells after AA treatment and found that after 15 min of treatment with 40 mM AA, only 33.2% of the wt cells survived in wt strain (Figure 5B), most likely due to the inability to restore pH_c_ homeostasis. By contrast, the survival of the *ck1*Δ cells was approximately double at this time point. Even after 60 min of AA treatment, more than half (54.8%) of the *ck1*Δ cells survived compared to only 6.8% and 11.2% in the wt and the complemented strain, respectively (Figure 5B). This suggests that the increased plasma membrane H^+^-ATPase activity of the *ck1*Δ mutant makes it more resistant to AA-triggered cell death despite its lower homeostatic pH compared with the wt strain.

### 3.5. Ck1 Controls Hyphal Chemotropism towards Acid pH

Previous studies revealed that germ tubes exhibit chemotropism towards tomato root exudate [30] or acidic pH [43]. Here we found that the wt and the complemented strains showed chemotropic growth towards acid pH, whereas the *ck1*Δ mutant grew preferentially toward alkaline pH (Figure 6). Thus, Ck1 is required for acid pH-triggered hyphal chemotropism of *F. oxysporum*.

### 3.6. Ck1 Positively Regulates the MAPK Fmk1

Alkalinization was previously shown to trigger phosphorylation of the conserved mitogen-activated protein kinase (MAPK) Fmk1, which is required for invasive growth and pathogenicity [6], whereas extra- or intracellular acidification caused a decrease of Fmk1 phosphorylation with a concomitant increase in Mpk1 phosphorylation [43]. Here we compared the phosphorylation levels of Fmk1 and Mpk1 in the wt and the *ck1*Δ mutant before and after the addition of diethylstilbestrol (DES), a specific inhibitor of the H^+^-ATPase Pma1, which causes a rapid acidification of pH_c_ [43]. We found that in mycelium grown at pH 6.0, the basal levels of phosphorylation of Fmk1 and Mpk1 were markedly reduced in the *ck1*Δ mutant compared to the wt and the complemented strain (Figure 7, time 0′). Furthermore, as previously reported [43] addition of the specific Pma1 inhibitor DES caused a rapid increase of Mpk1 phosphorylation and a decrease of Fmk1 phosphorylation levels, but no such response was detected in the *ck1*Δ mutant (Figure 7, times 5′ and 10′).

### 3.7. Ck1 Is Essential for Invasive Hyphal Growth and Pathogenicity of F. oxysporum

Previous work established that an increase in ambient pH promotes infection-related functions in *F. oxysporum,* including invasive hyphal growth, a process defined as the capacity to cross a cellophane membrane on a plate [6]. As previously reported, the wt, the ectopic transformant, and the complemented strain were able to penetrate the cellophane both on unbuffered MMA plates as well as on plates buffered at pH 7, but not on plates buffered at pH 5 (Figure 8A and Appendix A). However, the two *ck1*Δ mutants were unable to penetrate the cellophane membrane in any of those conditions, suggesting that invasive hyphal growth of *F. oxysporum* is strictly dependent on Ck1.

Invasive hyphal growth is a prerequisite for plant infection [6]. We found that tomato plants inoculated with conidia of the *ck1*Δ mutants were still alive after the infection experiment, whereas mortality rates in plants inoculated with the wt, the ectopic transformant, or the complemented strain were close to 100% (Figure 8B and Appendix A). Thus, Ck1 is essential for the pathogenicity of *F. oxysporum* on tomato plants.

## 4. Discussion

Casein kinases have multiple roles in fungi. So far, the role of casein kinase in fungal pathogenicity on plants has not been studied in detail. Here we generated and characterized knockout mutants in the single casein kinase homolog *ck1* of *F. oxysporum.* The results of our study support a role of Ck1 as a key regulator of intra- and extracellular pH. In contrast to *S. cerevisiae,* where the *yck1 yck2* double knockout is lethal, the *ck1*Δ mutants in *F. oxysporum* were viable, although they exhibited a strongly reduced colony growth rate and reduced conidiation. A similar restricted colony growth phenotype was reported in *ck1*Δ mutants of the wheat scab pathogen *Fusarium graminearum* and the human pathogen *C. neoformans* [26,28]. Meanwhile, a *yck2*Δ*/yck2*Δ strain of *C. albicans* was defective in hyphal growth under inducing conditions and instead formed chains of elongated yeast cells that resembled pseudohyphae [24]. Similar to *F. oxysporum*, a severe conidiation defect was also observed in the *ck1*Δ mutants of *M. oryzae* [27,44], and *F. graminearum* [28]. Taken together, these findings suggest that Ck1 plays a broadly conserved role in polar hyphal extension, particularly on solid media, as well as in the formation of conidia.

We noted that the *F. oxysporum ck1*Δ mutant showed increased sensitivity to SDS and hyperosmotic stress. In a previous study, *C. neoformans* Ck1 was suggested to regulate the expression and phosphorylation levels of the MAPKs Mpk1 and Hog1 under cell wall and hyperosmotic stress conditions, respectively [26]. Furthermore, the *C. albicans ck2*Δ mutant was also hypersensitive to SDS, but in contrast to *F. oxysporum*; it also displayed increased sensitivity to the cell wall perturbing agents CFW and CR [24]. This suggests a general role of Ck1 in stress responses, although the quantitative contribution to individual types of stress appears to differ between different fungal species.

Here we establish that Ck1 negatively regulates the H^+^-ATPase activity of Pma1 in *F. oxysporum*. This is in line with previous studies in yeast, where Pma1 activity increased 3-fold in starved cells of a Ck1-deficient strain [16]. In *S. cerevisiae*, negative regulation of Pma1 by Ck1 was shown to function via phosphorylation of the Ser-507 residue [16]. Importantly, the Pma1 proteins of *F. oxysporum* and other filamentous fungi contain a conserved Thr residue at the analogous position together with a casein kinase consensus phosphorylation motif [40,41], suggesting that negative regulation of Pma1 by Ck1 is conserved in fungi.

Interestingly, the extracellular alkalinization observed during colony growth of *F. oxysporum* wt was abolished in the *ck1*Δ mutant, most likely because of the increased H^+^-ATPase activity. Another consequence of Pma1 hyperactivation is the higher resistance to AA-induced cell death observed in the *ck1*Δ mutant, similar to that described in *S. cerevisiae* [45], where dissociation of AA inside the cell leads to a decrease of pH_c_ and triggers programmed cell death [46].

In spite of the increased Pma1 activity, the homeostatic pH_c_ in the *ck1*Δ mutant was significantly lower than in the wt strain. This unexpected finding could be due to a compensatory effect of other ATPases, such as the vacuolar ATPase, whose activity was shown to be regulated in response to cytosolic and ambient pH in coordination with Pma1 [8,47]. In addition, the activity of Pma1 as an electrogenic pump is strongly dependent on other plasma membrane transporters, including the K^+^ transporters Trk1 and Trk2 [48,49]. Alterations in the expression and activity of these transporters may thus affect pH_c_ homeostasis in the *ck1*Δ mutant.

A critical step during the infection process of *F. oxysporum* is the capacity to locate roots in the soil, which relies on the chemotropic sensing of peroxidase enzymes released by plant roots [30]. Furthermore, it was recently shown that germ tubes can sense and re-direct growth towards a gradient of acid pH, and that this growth polarity is inverted towards alkali in a mutant lacking the MAPK Mpk1 [43]. Here we found that *ck1*Δ mutants also exhibit inverted chemotropism towards alkali rather than towards acidic pH. This abnormal pH tropism could be related to the increased Pma1 activity in this mutant, which leads to acidification of the hyphal surroundings and could thus interfere with Mpk1-mediated sensing of the acid gradient. Further supporting this idea, we found that regulation of Mpk1 phosphorylation was impaired in the *ck1*Δ mutant. Importantly, phosphorylation of the MAPK Fmk1, which controls invasive growth and pathogenicity on plants, was markedly reduced in the *ck1*Δ mutants, which were also impaired in their capacity to penetrate cellophane membranes and to cause disease in tomato plants. These findings are in line with a recent study showing that acidification of the rhizosphere by the endophytic rhizobacterium *Rahnella aquatilis* through secretion of gluconic acid (GlcA) inhibits plant infection by *F. oxysporum* [50]. Thus, Ck1 has a conserved role during infection by different types of fungal pathogens such as *C. neoformans* [26], *M. oryzae* [27] and *F. graminearum* [28].

Taken together, the results of our study shed new light on the role of Ck1, a negative regulator of Pma1. Further studies are required to fully elucidate the complex regulation of this essential plasma membrane H^+^-ATPase, which involves additional protein kinases such as the activators Ptk2 and Hrk1 [14,15]. Understanding pH homeostasis will most certainly reveal new ways to control fungal growth, development, and pathogenicity.

## Figures and Tables

**Figure 1 jof-08-01300-f001:**
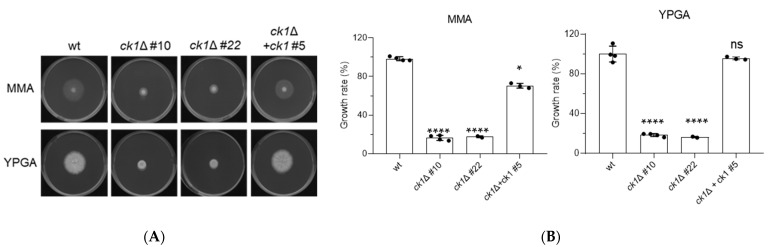
Loss of *ck1* strongly impacts colony growth of *F. oxysporum*. (**A**) Aliquots of 5 × 10^4^ fresh microconidia of the indicated strains were spot-inoculated on minimal medium agar (MMA) or complete medium agar (YPDA) plates. Colonies were imaged 3 days after inoculation. The plates shown are representative of three independent experiments with three plates each. (**B**) The increase in colony area was calculated after spot-inoculation of the indicated strains on MMA and YPGA and normalized to that of the wt strain (100%). *p* < 0.05 (*), *p* < 0.0001 (****) and not significant (ns) versus wt, according to Welch’s *t*-test. versus wt, according to Welch’s *t*-test. Data show the mean and standard deviation from at least three independent experiments.

**Figure 2 jof-08-01300-f002:**
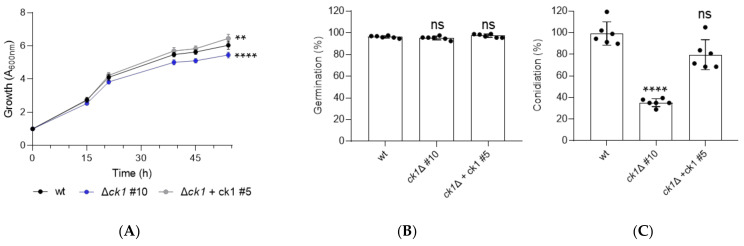
Loss of *ck1* severely affects growth and conidiation in a liquid medium. (**A**) Growth of the indicated strains was monitored in microwell plates containing liquid yeast extract-dextrose (YD) medium by measuring absorbance at 600 nm. Values were normalized to time 0. *p* < 0.01 (**) and *p* < 0.0001 (****), ns, not significant versus wt, according to Welch’s *t*-test. Data presented are the mean ±and standard deviation of three replicate microwells from one representative experiment. The experiment was performed twice with similar results. (**B**) The percentage of germinated microconidia was determined 15 h after inoculation in PDB medium. ns, not significant versus wt, according to Welch’s *t*-test. The data presented are the mean and standard deviation of two biological replicates with three technical replicates each. (**C**) Microconidia production by the indicated strains was determined after 48 h growth in liquid PDB medium and normalized to the wt strain (100%). *p* < 0.0001 (****) versus wt, according to Welch’s *t*-test. The data presented are the mean and standard deviation of two independent experiments with three technical replicates each.

**Figure 3 jof-08-01300-f003:**
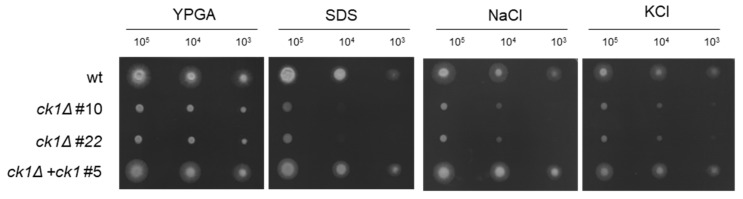
Ck1 contributes to the response to membrane and hyperosmotic stress. Aliquots of 10^5^, 10^4^, and 10^3^ fresh microconidia of the indicated strains were spot-inoculated on YPGA plates in the absence or presence of 0.075% (*w/v*) SDS, 1.2 M NaCl, or 1.2 M KCl. The plates were imaged after two days of incubation at 28 °C. Images are from one representative experiment. Experiments were performed twice, each with three independent plates per growth condition.

**Figure 4 jof-08-01300-f004:**
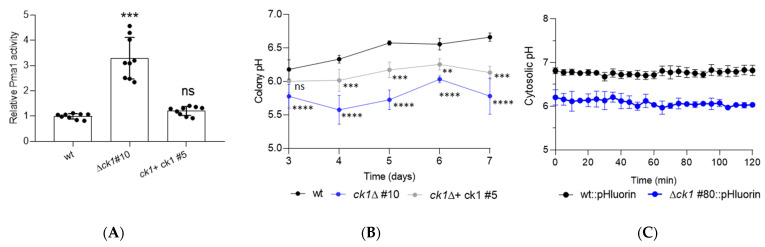
Ck1 negatively regulates Pma1 activity and controls pH homeostasis. (**A**) Pma1 ATPase activity was assayed in total membrane fractions isolated from germlings of the indicated strains after 15 h incubation (see Section 2). The activity was normalized to that of the wt strain. *p* < 0.001 (***), ns, not significant versus wt according to Welch’s test. Data shown are the mean and standard deviation from three independent experiments with three technical replicates each. (**B**) Aliquots of 5 × 10^4^ fresh microconidia of the indicated strains were spot-inoculated on minimal medium agar (MMA) plates and incubated for 7 days. At the indicated time points, 4 mm^2^ squares were cut from the center of each colony, homogenized in 50 µL of ultrapure water, and the pH was measured with a microelectrode. *p* < 0.0001(****), *p* < 0.001 (***), and *p* < 0.01 (**) versus wt according to two-way ANOVA and Turkey Test. Data presented are the mean and standard deviation from three independent biological experiments, each with three colonies analyzed. (**C**) Cytosolic pH (pH_c_) was monitored in germlings of the indicated strains expressing the ratiometric pH probe pHluorin Microconidia were germinated 15 h in YD medium buffered at pH 7.4, germlings were washed and resuspended in KSU buffer at pH 6.0, aliquoted into microwells and pre-incubated 60 min at 28 °C, before monitoring pH_c_ spectrophluorometrically every 5 min for 2 h. For pH_c_ measurements, the ratio between the emission intensities (collected at 510 nm) after excitation at 395 nm and 475 nm was calculated. Data show the mean and standard deviation of three independent replicate wells from one representative experiment. Experiments were performed three times with similar results.

**Figure 5 jof-08-01300-f005:**
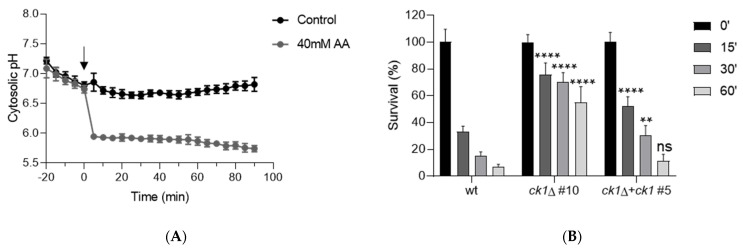
The *ck1*Δ mutant displays increased resistance to acetic acid-induced cell death. (**A**) The pH_c_ was monitored in *F. oxysporum* wt germlings expressing pHluorin. Microconidia were germinated during 15 h in liquid PDB, germlings were washed and transferred to KSU buffered at pH 3.0, aliquoted into microwells and pre-incubated 60 min at 28 °C before adding or not 40 mM acetic acid (AA). Samples were monitored spectrophluorometrically every 5 min starting 20 min before AA addition. pH_c_ was calculated as the ratio between the emission intensities at 510 nm after excitation at 395 nm and 475 nm and normalized to the standard curve. Data show the mean and standard deviation from three replicate microwells from one representative experiment. Experiments were performed at least twice with similar results. (**B**) Microconidia of the indicated strains were germinated during 15 h in PDB, then the pH of the medium was adjusted to 3.0 with diluted HCl and germlings were pre-incubated for 1 h at 28 °C before adding 0.4 mM AA. Germlings collected before (time 0) or at the indicated times after AA addition (min) were diluted and plated on complete medium plates. After two days of incubation at 28 °C, the number of colonies was counted, and survival was calculated. *p* < 0.0001(****), *p* < 0.001 (**), ns, not significant versus wt according to two-way ANOVA and Bonferroni Test. Bars show the mean and standard deviation from three independent biological experiments.

**Figure 6 jof-08-01300-f006:**
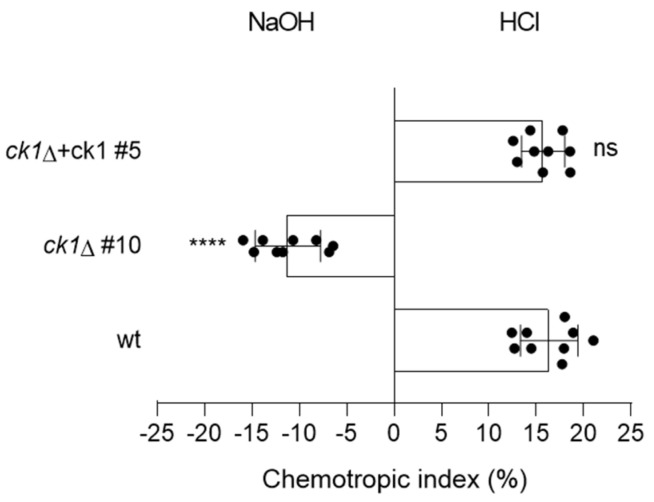
Ck1 is required for the chemotropic response to an acid pH gradient. The directed growth of germ tubes of the indicated *F. oxysporum* strains was determined after 8 h exposure to opposing gradients of 25 mM HCl and NaOH. *p* < 0.0001 (****), ns, not significant versus wt according to Welch’s *t*-test. Data show the mean and standard deviation from three independent experiments with three replicates (n = 500 germ tubes per experiment).

**Figure 7 jof-08-01300-f007:**
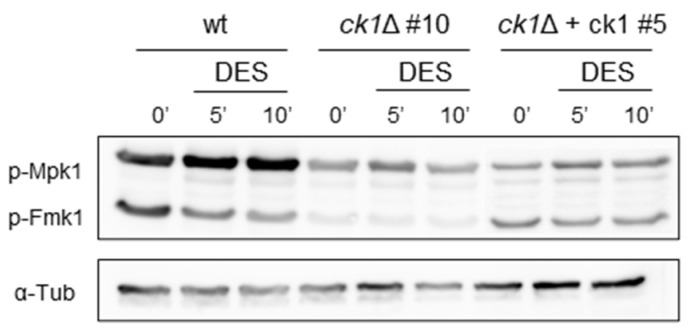
The *ck1*Δ mutant shows altered MAPK phosphorylation levels. Microconidia of the indicated *F. oxysporum* strains were germinated in PDB for 15 h at 28 °C; then germlings were washed and resuspended in KSU buffer, pH 6.0. After 1 h at 28 °C, 0.5 mM of the specific Pma1 inhibitor diethylstilbestrol (DES) was added. Protein extracts collected either before (time 0) or at the indicated times after DES addition, were subjected to immunoblot analysis with anti-phospho-p44/42 MAPK antibody, which specifically detects the phosphorylated version of the Mpk1 and Fmk1 MAPKs. Anti-α-tubulin (α-Tub) antibody was used as a loading control.

**Figure 8 jof-08-01300-f008:**
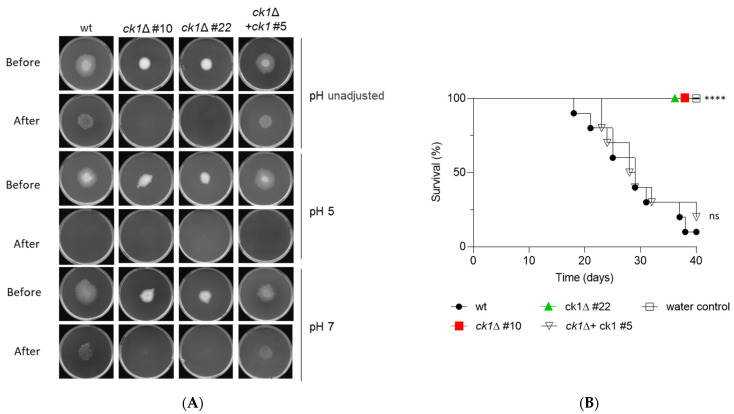
Ck1 is required for invasive hyphal growth and pathogenicity on tomato plants. (**A**) Aliquots of 5 × 10^4^ fresh microconidia of the indicated strains were spot-inoculated on top of cellophane membranes placed on MMA plates that were either unbuffered of buffered to pH 7.0 or 5.0 with 100 mM MES. Plates were imaged after 3 days incubation at 28 °C (before), then the cellophane membrane with the fungal colony was removed, and plates were incubated for an additional day to visualize the presence of mycelium on the plate, indicative of penetration through the cellophane (after). The images shown are representative of three independent experiments, each with three plates per treatment. (**B**) Kaplan–Meier plots showing the survival of tomato plants (cv. Moneymaker) inoculated by dipping roots into a suspension of 5 × 10^6^ fresh microconidia/mL of the indicated fungal strains. Survival of tomato plants was recorded for 40 days. Ten plants were used per treatment. *p* < 0.0001 (****) versus wt according to the log-rank test. The data shown are from one representative experiment. Experiments were performed three times with similar results.

## Data Availability

Not applicable.

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
