# Peer review of "Fusarium oxysporum Casein Kinase 1, a Negative Regulator of the Plasma Membrane H+-ATPase Pma1, Is Required for Development and Pathogenicity"

_jof, 2022, doi:10.3390/jof8121300_

Round 1

Reviewer 1 Report

The manuscript by Mariscal et al. investigates and characterises the function of a casein kinase-1 (Ck1) in Fusarium oxysporum. The authors hypothesised and provided evidence that Ck1 is involved in the regulation of cytosolic pH in Fo and alkalinisation of its surrounding environment. The authors also show evidence that Ck1, and by extension the ability to regulate pH, is required for full virulence towards tomato. 

Generally the manuscript is well-written and the experiments include appropriate controls. A few formatting errors are highlighted in the manuscript for correction.

As a minor comment, some experiments show the results of two independent KO mutants while other experiments only show results of a single KO mutant. If the data is available for both mutants, for all experiments, it would be good to include the data (or provide as a supplementary). It would also be recommended to present the data for an ectopic mutant expressing the KO cassette to demonstrate that results from the experiments are specific to the Ck1 KO mutation, and not the expression of the cassette. It would also be good to include some images of the plants from the infection studies (even as supplemental images). With reference to image 8B, it is difficult to easily identify and interpret the survival rates of the different treatment groups. It would be advantageous to the readers for this data to be presented in a different graphical form.

While the discussion is well-rounded and relates each result back to the appropriate literature, there is little discussion of knowledge extension or "what's next". In terms of pH regulation in filamentous fungi, what other questions are there and how does the Ck1 pathway relate to those questions (e.g. the interaction between Ptk2 and Ck1 as mentioned line 51 of intro)?

Reviewer 2 Report

The authors of “Fusarium oxysporum casein kinase 1 (Ck1), a negative regulator of the plasma membrane H+-ATPase Pma1, is required for development and pathogenicity” presented an extremely focused study describing the role of Ck1 in F. oxysporum development and pathogenicity. In root-infecting vascular pathogenic fungi, F. oxysporum induces an increase in the pH of the surrounding host tissue. Alkalinization is supposed to related to the role of cytosolic pH (pHc), which is mainly controlled by the essential plasma membrane H+-ATPase Pma1. It is noteworthy that the authors created mutants of F. oxysporum lacking Ck1, the negative regulator of Pma1, and assessed Pma1 activity, alkalinization of the surrounding medium and pathogenicity on host plants using ck1Δ mutants. All the data supports the hypothesis that Ck1 acts as a key regulator of development and virulence of F. oxysporum.

Some suggestions:

Line 264-269: For ck1 deletion, how many right ck1∆ mutants were isolated from the  transformants? What was the knockout rate?

Line 272: Figure S2D, is that right? No Figure S2D in supplementary data.

Figure S1B: “6,1kb” should be” 6.1kb” or “6,100bp”. Similarly, “3,4kb”.

Figure S1C: “1,0kb” should be” 1.0kb”.

Figure S1D: “ck1Δ pHluoin” should be “ck1Δ pHluorin”. “4,1kb” should be” 4.1kb”. Similarly, “2,5kb”.

Line 307-308: Figure S2 is not related to CFW or CR.

Line 430: The mycelium penetrating through the cellophana in Figure 8A can be described as “penetrating hyphae”, not “invasive hyphae”. So, ”invasive hyphal growth” in Line 430 is not applicable.

References section: Only the first letter of the article title should be capitalized, others should be lowercase. Please pay attention to reference 3, 32, 35 and 44. Meanwhile, abbreviations for magazine names, such as reference 2, 4,6 and 9.

Reviewer 3 Report

            In this manuscript, the authors address the importance of limiting extracellular acidification by the proton pump Pma1 via casein kinase inhibition in the physiology and pathogenicity of the pathogenic plant fungus F. oxysporum.  Specifically, the authors provide evidence that knockout of the single casein kinase in this fungus results in activation of Pma1, extracellular acidification, altered chemotropism, and ultimately reduced pathogenicity.  Overall, this is a carefully done and interesting study that extends from general properties of the casein kinase knockout to a more specific focus on intracellular and extracellular pH homeostasis to invasive properties and pathogenicity.  I would recommend only that a few minor points be clarified.

1.  In Figure 4, it makes sense that the colony pH is reduced in the ck1∆ mutant, given the increased activity of Pma1.  However, it is less clear why cytosolic pH is reduced in this mutant in Figure 4C.  If the cells were maintained a low extracellular pH, that could contribute to low cytosolic pH, but the both wild-type and mutant germlings are washed and placed in pH 6 buffer before cytosolic pH measurement is begun, and there is no decrease over time.  If Pma1 is more active, it might be expected that cytosolic pH would be higher than in wild-type.  This experiment requires further clarification and explanation.

2.  In Figure 5, the authors show that the ck1∆ mutant is more tolerant of acetic acid stress and they argue that this is because of increased export of acid.  However, the previous figure shows that cytosolic pH is actually lower in the mutant.  It would be interesting to show the cytosolic pH response of the ck1∆ mutant to acetic acid at low pH in Figure 5A along with the wild-type result, if possible.  If that is not possible, the authors need to explain the apparent conflict between Figures 4 and 5.

Minor comment:  The wording sentence on Lines 497-500 is not very clear, though the overall point is made.

Reviewer 4 Report

This paper describes a role of casein kinase I in the negative regulation of the plasma membrane H+-ATPase (Pma1) in Fusarium oxysporum. The authors generated casein kinase I mutant (ck1) strains and found the mutation reduces cell growth both in liquid medium and on plate medium, increases sensitivity to SDS treatment and hyperosmotic stress, increases Pma1 activity and perturbs both intracellular and extracellular pH homeostasis, increases resistance to acetic acid stress, affects pH chemotropism, reduces Fmk1 phosphorylation, and negatively impacts on invasive hyphal growth and pathogenicity on tomato plants. The paper is well-written; the experiments were nicely executed; the conclusions are sound, and statistical analysis is rigorous. Although mutant analysis of the putative Ck1 phosphorylation site in Pma1 might enhance the paper, it will not significantly change conclusions of this study. I have a few minor comments.

1. Figure S1B: the 3.4kbp marker bar needs to move up a little to align with the bands on Southern blot.

2. Figure 1B legend: “the wt, the and 11” should be “wt, the ck1 deletion mutant, and 11”

3. Line 99: “restriction enzymes PstI/NsiI” should be “restriction enzyme PstI or NsiI”.

4. Lines 103-110: How does the wild ck1 gene integration work? I don’t see how co-transformation of the PCR products of wild-type ck1 gene and phleomycin resistance cassette would lead to ck1 integration by selecting Phl+ clones. For this purpose, in studies using S. cerevisiae strains, we generate a fusion PCR product with the selection marker flanked by yeast chromosomal DNA sequences that would lead to a two-site recombination. If this is a commonly used method in the F. oxysporum field, a citation would be sufficient.

5. Line 266: “several ck1 deletion” mutants should be “two ck1 deletion mutants (#10 and #22 isolates)”.

6. Line 272: Figure S2D does not exist.

7. Line 291: Exponential growth in Figure 2A is not evident. Could “exponential growth” be revised to “growth in liquid medium”? Leave as is if this kind of growth curve is considered exponential in the authors’ research field.

8. The most surprising observation is presented in Figure 4C. One would expect cytosolic pH would increase when Pma1 is more active and pumps more protons out of the cytoplasm. Could authors speculate on how ck1 mutants have reduced cytosolic pH? Are there published results on reduced cytosolic pH when plasma membrane H+-ATPase is more active?
